# The Diaconal Work of Sisters Kristína and Mária Royová—An Example of the Link between Christian Anthropology and Social Work

Peter Jusko [1], Albín Masarik [2,*] and Ján Nvota [3]

1 Department of Social Work, Faculty of Education, Matej Bel University, Ružová 13, 974 11 Banská Bystrica, Slovakia; peter.jusko@umb.sk

2 Department of Theology and Religious Education, Faculty of Education, Matej Bel University, Ružová 13, 974 11 Banská Bystrica, Slovakia

3 Republic of Serbia, Autonomous Province of Vojvodina, Provincial Secretariat for Education, Regulations, Administration and National Minorities—National Communities, Bul. Mihaila Pupina 16, 21101 Novi Sad, Serbia; nota@stcable.net

* Correspondence: albin.masarik@umb.sk

**Abstract:** Considering the intermingling of problems in today's multi-crisis environment, this text explores the possibilities of intertwining social work and pastoral care. In the search for effective approaches, we find dynamic patterns in the activities of the Royová (Roy) sisters. Their diaconal work is an important source for social work history illustrating how social work took the form of diaconal (charity) work with a rich pastoral reach at the time. Their activities represent a natural link between Christian anthropology and social work. This study mainly investigates the Christian (spiritual) basis of the social and charitable activities of the Royová sisters, the beginnings of the institutionalisation of social and charitable work in Slovakia and Serbia through the organisations founded by the Royová sisters, the Christian-social interpersonal contribution of the Roy sisters to the development of Slovak and European social work personified by their cooperation with several personalities of social and charitable work at the international, national, and local levels, and the contribution of the Roy sisters in the creation of women's, volunteer, and international roots of social and charitable work in Slovakia and Europe. In their responses to the needs of their environment, we find significant stimuli for pastoral theology, which is supposed to respond to the needs of the multi-crisis environment of today.

**Keywords:** Kristína Roy (1860–1936); Mária Roy (1858–1924); social work; pastoral theology; Christian anthropology; diaconia; church; history of social work

## 1. Introduction

Today's people live in an era in which one crisis overlaps another. Humanity has not yet recovered from the COVID-19 pandemic and its consequences, and the Russian invasion of Ukraine created a security crisis and the related energy and subsequently economic crisis situation. At the same time, we are observing the widespread onset of negative and irreversible processes in ecology. This is a multi-crisis environment and one has to cope with its effects on one's existence. The crippling breadth of the problems is reinforced by the awareness of the limited sources of effective help. The question is therefore to what extent the helping professions and churches in the world can respond to the problems of such a radically changed environment.

We find a valuable example of effective pastoral and diaconal (social) help in the activities of the Roy sisters, Kristína and Mária. Their story is an important source of effective forms of church pastoral and diaconal help and also for today's social work in its secular profile. In academic disciplines, we can make an exact distinction between

social work, diaconia, and pastoral theology. But in practice, these disciplines significantly overlap and take the form of support and assistance to a specific client.

The terminological basis of our historical research will be the historical and contemporary understanding of the broad perception of risks and unfulfilled material and spiritual needs of people and the response to them in the work of the Roy sisters. In this context, we follow their activities in relation to their faith and Christian anthropology.

The development of social work in Europe has not been smooth. In the historical study of a specific phenomenon (the social and charitable activities of the sisters Kristína and Mária Roy), it is necessary to rely on the understanding of social work in a given historical period. The Christian tradition of social work based on love of neighbour forms the contemporary and national starting point for understanding their social and charitable activities. Charitable work belongs as an independent theological discipline to practical theology. It has many features in common with social work, but according to the New Testament, differs from it in its origin, motivation, and purpose, as diaconal work starts from the spiritual dimension and communicates it non-verbally to the addressees of its activity.

The social, charitable, and pastoral activities of the Roy sisters represent a practical performance of social work based on Christian anthropology, thus co-creating an important source of history not only of Slovak but also of European social work. According to the International Federation of Social Workers (Global Definition of Social Work 2014), social work is currently perceived as a practical profession and a scientific discipline that promotes social change, social cohesion, human rights, and freedom, involving people and structures in solving life problems and strengthening social welfare. Christian anthropology is a branch of systematic theology that deals with a person from the point of view of Christian theology, especially the existence of a human being and its destiny before God, while serving for spiritual and ethical orientation in a life based on the Christian faith.

The creation account (Gen 1-2) already shows that man is a human being, created in the image and likeness of God, given the capacity to "rule over" creation (Gen 1:26). The statement "it is not good for man to be alone" (Gen 2:18a) expresses that man is a being intended for closer and wider social relations (Gen 1:28). The fact that all humans reflect God's image carries in itself an inherent dignity.

Hanes (2022, p. 26) states, "From the biblical theological point of view, a human being is a whole that can be viewed from either material ("dust") or spiritual ("soul")". In a footnote, he explains that "In Genesis 2:6 the created human is called both "dust from the ground" and "a living soul". There is no preposition "from" (the dust) in the original Hebrew". Scripture does not describe man in today's language of psychology. Hanes (2022, p. 26) rightly notes that "in the Bible emotions, reason and will are lumped together under the notion of "heart"—mind and logic are not isolated from emotions and passions".

Both the OT and the NT emphasise that man has deformed his relationship with the Creator, and this deformation also has negative and deforming effects on his social being and the fulfillment of his task of ruling over creation. Despite this, or precisely because of this, the New Testament shows man as an object of God's love and salvation in Christ's sacrifice. The believer thus receives a new perspective that goes beyond his earthly existence.

The literary work and the songwriting of the Roy sisters, as well as their whole diaconal work, provide insight into their understanding of Christian anthropology. In this article we explore their songs included in the hymnal of four hundred Christian Songs, published by the Baptist Union in Slovakia in 2014 (*Spevník—400 kresťanských piesní 2014*). It contains seventy-six songs in which the Roy sisters participated as authors of the text and/or of the music, some of which are translated from other languages. We focus only on their original work, which represents the content expressed and formulated by them.

These songs describe man as a human being, created by God. Song No. 17, v1 speaks about God. The following refers to their Christian anthropology in relation to God: "He is the foundation of being . . . where he is not, there is no life. . ." (and similarly in v2, "where he is not, there is no joy.").

They perceive the person before God as a sinner. He or she needs their sins to be forgiven and be granted salvation. But because of God's goodness, his/her sinfulness does not lead the person to despair. Song No.19 repeatedly expresses that God is love. The first verse (hereafter v1) says that the person—the sinner—should hear this, should "wake up from their sleep" and "seek their salvation". At the same time, sinners should learn that "eternal salvation is only in Him". This expresses God's exclusive claim on man, and that man can only find in Him "the source of peace, tranquillity", and "in Him is all the meaning of life". The whole song is repeatedly penetrated by the expression "God is love" (in each of the 3 verses, it is repeated 3 times in the text and again 3 times in the chorus, that is, 18 times in total). It carries the message that God loves man. A similar statement about Christ is in the song No. 44, v4: "He came down to earth out of love, to redeem us from our sins".

That is why Roy speaks to the sinner in song No. 113: "Return, child, from the way of error, sin will destroy you before you know it! The world is a dangerous desert, without a leader you will perish in it". Or in song No. 113, v4, the song calls, "Do you feel unworthiness and guilt? Do not give in, I am afraid. Come to the cross of Jesus, salvation awaits you there". This call is urgent because of the vulnerability of the human being. That is why song No. 297 v4 calls out, "Make peace with God. You are like grass. You can die suddenly. . . . make peace with God".

The missionary call of the Roy sisters to repentance is not a matter of creating fear in sinners, but it is a joyful invitation to fellowship with a loving God. At the same time, it does not focus only on solving the problems of everyday life but extends into eternity. We find it in song No. 8, v1: "Hallelujah, rejoice mankind, God has given us his Son. He lovingly calls sinners to his homeland, where he has prepared for them infinite joy". Chorus "He loves us, He loves us, he loves people with eternal love. . ." Similarly, song No. 130, v3, expresses, "The Lord calls you; He waits for you, He wants to give you the glory of heaven. Make up your mind, do not delay, because you will regret it!" The missionary call in these songs also has a pastoral dimension. Song No. 13, in v3, comes to the view of a person whose relationship with God will show a way out of difficulties. The sinner is to surrender his heart to Christ, and the needy and sick should also go to Him, because "His wounds heal you, the heavenly oil He has for you, He has prepared glory for all". From faith in God, here grows the hope of improving the condition of the addressed person (forgiveness, healing, a new outlook), and this approach was also the driving force behind the mission, diaconal (social) and catechetical activities of the Roy sisters.

Their songwriting richly expresses the forgiveness of sins as a fact about which the Christian can already know. Song No. 113 expresses, "Sins are already forgiven, no condemnation awaits you. Let your poor heart not be afraid of the death penalty!" Likewise, in song No. 68, "Don't forget, don't forget that Jesus bought you dearly! To reconcile you with the Father, and gain you eternal bliss". Likewise, in v3, the song calls out, "Don't forget. . . how much your soul cost, the sufferings, the humiliation of Golgotha, the heavenly King! To snatch you from power of hell; from His wounds the blood on the shameful cross flowed". By accepting forgiveness, a person undergoes a real change, which is expressed by song No. 44 in v4: "With a pure heart, let us all sing his praises together".

This collection of songs bears witness that Christian spirituality provides an answer to man's inner needs, like in song No. 15: "When you are thirsty, rush to the source, which springs from the rock. The risen Jesus is the source of pure living water". The song reflects the inner needs of the person who feels exhausted in difficult social struggles. In this context, the chorus is fascinating and valuable: "God in love embraces us, glory, honour, hallelujah". By singing this song, a Christian learns to perceive and experience God's love, and as it is expressed in song No. 389, v1, the Christian learns that "He heals wounded hearts, strengthens weary souls. In the heavenly glory, my Saviour, the consolation, the eternal strength is the source". V2 also added the words, "the living source of happiness and light". Therefore, song No. 24, v1 claims "Only he lives whom you revive with

your love". The relationship between God's love and the quality of human existence is expressed here.

That is why a person who internally opens up to God can experience the joy of faith and the new position before God. Song No. 172, v1 formulates this: "Let our hearts be filled with joy, our souls burn with gratitude. The Father in heaven calls us children, we have reason to be happy". The Roy Sisters knew people well and that is why they also remind them in song No. 130, v2, of God's claim that their life should be unequivocal: "Jesus asks for the whole heart, because He gives the whole. There is no salvation for those who keep half to themselves". The emphasis on practical Christian life is also expressed in song No. 297, v1: "Do not lie to yourself" . . . You will not deceive God. . ." or in v2, "Don't live with pride" and shows its deforming consequences. Song No. 289, v1 also encourages perseverance in faith: "My brother, just don't stop believing! Before you know it, the Lord will come". Likewise, in v2, "How many wounds have you healed with love? My brother, do not stop loving!" Song No. 293 also expresses the desire for loyalty in the struggles of life: in v1, "Lord, grant that we may remain faithful, once before His golden throne. . . and in v2, "Let the malice of the world not frighten you, don't let the pain derail you. He who is afraid will not win. Glory does not wait for a coward". The chorus reminds us, "Stay faithful! . . . remember the eternal reward!" This loyalty is important because it expresses the goal of being a Christian (song No. 24, v4 reads, "To die for You, to serve You on earth is our work. To be with You forever enthroned in glory is our life's goal.") In admonishing Christians, they also use an eschatological perspective in song No. 291, v1: "Live beautifully in your early life, spread the fragrance of love around you. You wander where there are no more tears; there is no need for your condolences". Likewise, in v3, "Humbly carry the cross for Christ, the fire will clean your "Me". After fighting in white robes with Jesus, you will enter paradise".

A Christian can demonstrate himself even in the difficult circumstances of life. In song No. 233, v5, he/she realises that "The road is not always decorated with fragrant flowers, it is often covered with thorns. Even there the Lord leads his child because a rose blooms even among thorns". Song No. 255 asks the question in whom the Christian has support in various difficulties. In v1, there is the need to light the way and to offer a home "in a cold foreign land". In v2, he refers to the question of the "time of storms", the significant losses "when the happiness was burnt by the frost". In v3, it refers to the need to find strength to work "when there is no recognition", when going through "bitter tears of pain, disappointment", and hopeless situations "when no help is seen". In all these situations, the Christian finds support and strength in Jesus Christ.

Song No. 285, v2 presents Christianity as a life of service: "We did not come into the world to live only for our own well-being . . . we are to serve our neighbours . . . to spread God's glory. . ." In song No. 206, v1, this goal is Christocentrically narrowed: "To be allowed to live for Christ, to die for Him, oh that is a blessing beyond which there is none. It is worth suffering, fighting, carrying the cross for. For Him it is worth leaving this world". The relationship with Christ also refers to the conclusion of the Christian's earthly existence, as expressed by song No. 255, v4: "Who will open the gate for me when the pilgrimage is over? In whose hands shall I find the precious reward of faithfulness? Who will put me with love into the arms of the Father? Lord Jesus is the clear way to our eternal home". Song No. 238, v1 conveys a similar motive: "Let none of us forget what He has won for us. That we are freed from sins, destined for eternal glory, the glory that the throne awaits us". Also, Song No. 389, v3 shows this: "I have a fatherland in heavenly glory. I will go there when the Lord calls me". Likewise, in song No. 233, v6 in the form of a request, it expresses, "When the time approaches for me to finish my earthly run, and when the faintness of death comes upon me, grant me the joy to see the blissful shore!"

In the analysed set of songs, consciousness of God's judgment also appears. Song No. 24, v2 expresses, "You once stood above the horrors of the world of the flood; you will once again move the heavens and the foundations of the earth". The breath of Your mouth will destroy the wicked at once".

The Roy sisters were not indifferent to the fact that the sinful world will perish. It led them to their missionary work, and in song No. 113, v5, they call sinful people to repentance: "Come back, before the voice of God falls silent, His open arms will close! The gate of heaven will be closed forever in front of you". The need for a decision is also urged by song No. 107, v1—"suddenly the end of human days will come and the final judgment". The condemnation of those who do not repent is expressed in v6: "What a pain, to remain forever in the darkness, to perish in the depths!"

In summary, we can state that the Christian anthropology of the Roy sisters is deeply rooted in Christocentric spirituality, as confessed in song No. 13, v1 ("Jesus Christ, our King sits on the throne") and further developed in v2, "The Lord, the Triune God, He is the Ruler and no one else. ..."

The formulations in these songs of the Roy sisters are an expression of their Christian anthropology and were the base and theological starting point of their diaconal activities.

The Roy sisters lived and worked during a difficult period at the turn of the 20th century. Their legacy of selfless, active Christian work in literature, music, and the social area is valuable for both Slovaks and a wider audience. Their literary work and songwriting is part of their pastoral care and, together with their social work, represents complex care for people with problems that they cannot manage on their own.

This historical research aims to outline the social and charity work of the Roy sisters from Stará Turá, Slovakia, in the era of the Slovak National Revival (SNR). The SNR as a movement strived to prove that a separate Slovak nation with a specific identity exists. The Roy sisters were born and grew up in their local parish, where national activists held meetings which reflected in the sisters' work as well. They witnessed not only the political development, but also the troubles faced by the Slovak people at the time, mainly the parents and their children.

Therefore, these Christian sisters decided to put their faith in God into practice and focus on social and charitable work with the aim of improving living conditions and deepening the faith of the people they worked with.

Cultural development is also overcoming a significant change. In contrast to the Enlightenment rationalism of the previous period, it is strongly influenced by Romanticism, which strongly influences all areas of people's lives and influences the development of national consciousness.

In a multinational monarchy, this has caused serious complications. The process of national revival begins, because the issue of Czech, Hungarian, Croatian, Slovak, etc. arises. The national revival, the beginnings of which can be seen in the last years of the reign of Maria Theresa and especially Joseph II, is entering the second phase. The national history of social work investigates the development of social work, social security, theory and practice of social work in a geographically limited area, i.e., a country (Kováčiková 2000). This paper focuses on the historical research of Stará Turá, but also other parts of Slovakia, former Yugoslavia, and Europe affected by the social work performed by the Roy sisters. Brnula (2013, pp. 72–85) divides the development of social work in the Kingdom of Hungary (Slovakia was a part of it until 1918) into three periods:

1850–1900—Three social topics prevailed: poverty and homelessness, childcare, and alcohol addicts. Volunteers cooperated to establish poorhouses, shelters for abandoned and neglected children, and temperance associations (Blue Cross, operated by the Roy sisters). During this period, the legislation addressing social issues was formed.

1900–1914—The boom of social work in the Kingdom of Hungary and its gradual professionalisation. The differences between charity and social work were widely discussed. The first publications on social work were created. Organisations addressing the social issues were established.

In the Kingdom of Bohemia, Alice Masaryková founded the social approach, and the fight to enforce social measures began. She dedicated her life to social activism.

In Slovakia, the Roy sisters were active in the field (Brnula 2013).

1914–1918—the outbreak of WW1 halted industrial and economic progress, as well as the development of social work. Poverty ensued. As for social issues, mainly material assistance was addressed.

In the historical research on this topic, the findings were put into context and generalised to reconstruct the past based on the data and information obtained from the Museum of the Roy sisters in Stará Turá, its archives, interviews with the management, and valuable personal resources provided by Anna Chrťanová from Kysáč, Serbia[1].

Several types of historical sources were used in this study:

(a) Written sources. Besides official written sources, private sources were also used (private correspondence of the Roy sisters with their colleagues in Slovakia and abroad), written accounts of events (an extract from the interview with a former client of the Útulňa Children's Home, Milan Michalko, an interview with former deaconess Barbora Plesková, etc.), writings of Kristína Roy capturing the social situation and care provided at the time (e.g., her book entitled *Za presvedčenie—For conviction*).

(b) Material sources. A personal visit to the Museum of the Royová Sisters and the former Domov Bielych Hláv (seniors' home) in Stará Turá preserving the sisters' archives and personal items helped the authors to become acquainted with the specific socio-historical context.

(c) Visual sources. Valuable, mostly large-format photographs provided further information on the life and social care for people in the region of Stará Turá.

(d) Narrative sources. Accounts of people from Kysáč (former Yugoslavia), which the Roy sisters repeatedly visited, and their acquaintances Marta Roháčková and Jozef Roháček, the first translator of the Bible into Slovak.

In our paper, we will gradually identify the Christian (spiritual) basis of the Roy sisters' social and charitable activities in their lives and work, their collaboration with prominent personalities in social and charitable work at the international, national, and local levels, and their contribution to women, to voluntary and international roots of social and charitable work in Slovakia and Europe. By analysing these areas, we will try to identify the links between Christian anthropology and social work in the social and charitable activities of the sisters Kristína and Mária Roy.

## 2. The Lives of Kristína and Mária Roy

The life of the Roy sisters began against the background of the Slovak National Revival and the revolutionary years 1848–1849 in Europe and Hungary, during the search for Slovaks in the neo-absolutist Habsburg Monarchy (1859–1867) and the dualist monarchy (1867–1890) and sharpened national oppression in Hungary (1890–1914) and was completed after the establishment of the first Czechoslovak Republic after 1918 (Podolan and Viršanská 2014).

### 2.1. Family Background and Childhood

The Roy sisters were born at an Evangelical parish in Stará Turá (today the territory of Slovakia) to Evangelical pastor August Július Roy, a patriot and national activist and his wife Františka, born Holuby.

Mária, the older sister, was born on 26 November 1858 and Kristína on 18 August 1860. They had three other siblings.

Like many other spiritual leaders of the nation at the time, their father was an advisor and awareness raiser. He worked hard to raise the quality of life for people under the Javorina Mountains. During his era, the church and services were improved. After the old parish house burnt in 1852, he built a new one; he established new schools in Stará Turá and nearby solitary hillside settlements. In doing so, he set an example for his daughters that even construction projects can be implemented. He raised awareness selflessly and passionately, teaching the people economic basics as a significant part of his pastoral care for them.

Although both sisters were musically talented, Mária focused on music, while Kristína focused on writing. All children in the family were keen readers. The sisters did not find

housework a fulfilling career. They were faced with the local poverty and concerned about the lack of education (Slezáčková 1991, p. 7) and as Christians they also perceived the spiritual poverty of these people. Therefore, they decided to improve the situation. Their main concern was the lack of development for children. In an effort to improve the quality of the church life, they taught the children songs and Bible chapters.

*2.2. Spiritual Background*

After meeting preacher Ing. Kostomlatský in Czech Písek and missionary Dr. Baedeker from England (who worked in Siberia), the Roy sisters experienced a spiritual revival and fully embarked on their mission in Stará Turá. It seemed that the revival movement and Evangelical church were waking up "to informally build-up on the Slovak Pietism of the 18th century represented by Matthias Bel (1684–1749), *Magnum decus Hungariae*" (Great Ornament of Hungary) (Slavka 2007), who, "in all his works, showed that he was Christian, when he taught the Bible principles and the Ten Commandments as a priest, but also lived according to them as a father, teacher, and scholar. His national and religious tolerance provide an important example for our era and the space of the former Kingdom of Hungary" (Lupták 1999, p. 8). Therefore the motto of Pietism was: "Everything for the honour of God and general benefit" (Hanes and Hanesová 1999, p. 15).

The Roy sisters continued this tradition. On her 70th birthday (1930), Kristína Roy confessed, "*I only had two ideals in my life. Jesus Christ and the enslaved Slovak nation! To be able to spread the news of the glorious Saviour throughout the world, and help through Him only, the only medicine for the human souls—what a mercy for the Christian woman! And to be allowed to tell other nations about my tiny nation and convince them it is alive—what a bliss for the Slovak woman!*" (VeČERNICA Magazine, 1930). In this statement, the pastoral starting point of all their social and literary activity is clearly formulated. Jakš and Spišiak (2006) writes of Kristína Roy's four loves:

1.  **Love for God**—she (in)directly emphasised it in all her books, songs, poems, and musical compositions. She inserts it into dialogues between her protagonists who reflect her own mind. Instead of considering her love for God a part of religious or family traditions, she perceived it as a part of her personal faith, and it was the most important thing in her life: "Love the Lord your God with all your heart and with all your soul and with all your strength. (Deuteronomy, 6:5.)

2.  **Love for people**—in her works, she observes the human in social and spiritual need. Although the Roy sisters did not have opportunities like their slightly older contemporary from the Salvation Army, William Booth in England, whose foundation fed the poor for minimum fees night and day, the Roy sisters were concerned with the position of the poor, mainly abandoned orphans; therefore, they founded several social institutions to help the Slovak nation.

3.  **Love for the Bible**—she loved and extensively studied God's word. She read her Bible of Kralice (Czech translation) 49 times. She read it for the first time, as a child. For the 50th time, she read the Bible in Slovak, in Jozef Roháček's translation. She used her knowledge of the Bible in her literary and spiritual works (Jakš and Spišiak 2006, p. 6), inspiring many Slovaks from Evangelical families to read it.

4.  **Love for the nation**—her family environment was patriotic. The Stará Turá parish hosted many great, patriotic men who Roy describes with love and respect. On her patriotism, she has written, "We absorbed that devotion already with mother's milk". In her autobiography, she comments on the forced cancellation of Matica Slovenská (scientific and cultural institution) in 1875 and the supported grammar schools in Revúca (1874), Martin (1875), and Kláštor pod Znievom (1874) as follows: "*Suddenly, all grammar schools and Matica were taken from the Slovaks, yes, everything the poor people diligently built over that relatively short time. It hit my nation as a lethal blow. I remember when the Revúca students wandered to us in freezing winter, with frostbitten ears. The teachers who lost their income sought shelter with us. The persecuted writers, poets, and*

*editors of the banned magazines waited for our modest help*." (*Roy, K.*: 1992, p. 20.) Kristína Roy was no weaker in her love for the nation than any of its great, patriotic sons.

### 2.3. Kristína Roy, the Writer

Her first children's brochure entitled *Bez Boha na Svete (Without God in the World)* was published in 1893, and in 1902, it was translated into German and other languages (38 as of today). After these successes, Kristína Roy began writing fiction and poetry. She always loved children and wrote books for them. She founded the Literárne družstvo (Literary Cooperative) for like minds to create a common literary fund for publishing works and tracts.

Even 100 years later, Kristína Roy's books are still being published in Slovakia and abroad. During the period of totalitarianism in Slovakia, many of them were printed in Báčsky Petrovec (Yugoslavia).

### 2.4. Mária Roy, the Composer

Songs played an important role in the evangelisation work performed by the Roy sisters. *"If you don't wake up your soul by singing, it will not rise and come alive!"* (*Roy, K.,* 1992, p. 57).

One hundred years ago, when the Roy sisters wrote their spiritual songs in Slovak, no one had even dreamed of a Slovak evangelical songbook. Many believers around the world still use their songs (for example, in USA, or Serbia); they find spiritual support in them and are motivated by them to prosocial behaviour that grows out of faith in God.

## 3. The Spiritual, Social and Charity Work of the Roy Sisters

In her autobiography, Kristína Roy has written that after returning from Písek and Prague, the Lord sent her to bring spiritual and material help to her neighbours. Although the sisters had almost no property, as described in a number of publications, they trusted in God. They received the necessary means as his blessings and blessed with these gifts other people despite obstacles.

They performed their service of neighbourly love in Stará Turá and gradually founded four social charity institutions: Útulňa/Shelter (1880/1901), a hospital (1911), Chalúpka/The Cottage children's home (1926), and Domov bielych hláv/The White Heads' Home senior's home (1933). However, everything started with children.

### 3.1. Sunday School

The sisters could see the neglected children in the streets and decided to help them. They held meetings for them at their own home and later in a rented apartment to learn songs, which they translated and also composed.

Besides teaching songs, Kristína always told the children an interesting Bible story and together, they learned a short text by heart. The Roy sisters worked systematically and each child was taught the New Testament. These activities were an expression of pastoral care and were supposed to act preventively as protection against the problems faced by their social background.

Later, their parents would join the Sunday school Bible readings. Thus, the Sunday Evening Assembly for adults came to existence.

Their work with children brought visible progress. The previously neglected children roaming the streets as well as their parents learned about Christian values while receiving education.

### 3.2. Blue Cross

The Roy sisters developed their social and charity work under the umbrella organisation entitled *Modrý Kríž (Blue Cross)*, which was according Fischer (B. Fischer 1901, pp. 3–5), a temperance association, founded on 21 September 1877 in Geneva by Priest L.

L. Rochat and had its branches in Switzerland, Germany, France, Belgium, Denmark, the Austro-Hungarian Empire, Palestine, and other countries.

A branch of this association, which was based in Christianity, with the meetings opened with prayers, spiritual songs, and Biblical contemplations, was founded in Stará Turá on 1 January 1897 by Ján Chorvát (see Note 1 above), who later became the Roy sisters' brother-in-law. He married their youngest sister Božena (P. Brnula 2013, p. 75). Members of this organisation called one another brothers and sisters.

The Roy sisters founded a children's temperance club for their Sunday school students. Later, a temperance brotherhood was founded and joined exclusively by the believers.

Michal Slavka, (M. Slavka 2007) specified three basic goals of the Blue Cross Association (although not followed precisely):

1. Evangelisation
2. Church development
3. Social work

These three areas show the inseparable ties between the Roy sisters' social work and its rooting in Christian spirituality and church life.

### 3.3. Social and Charity Institutions and Associations

3.3.1. Shelter Children's Home

At the end of the 19th century, families living under the Javorina Mountains were poor due to the lack of job opportunities. Some of the inhabitants left for the USA to work but many people in Stará Turá travelled to sell goods door to door. From spring to autumn, the children's parents carried baskets of embroidery, lace, wooden tableware, haberdashery, and textiles on their backs to Austria or the more distant Sudetenland to sell it. They travelled and visited houses to offer their goods to feed their families. They left their small children at home with grandparents, relatives, or friends. However, this kind of care was often very poor. These children were "orphans with parents". The Roy sisters started training children in door-to-door selling.

For the care of such children, in 1880, the sisters bought a small house, renovated it, and accommodated children from several families. It was their first charity action, although it was not organised or based on legislation. This house was called Útulňa (Shelter). Later, unmarried deaconesses took over to run the house. At the time, they cared for 18 to 20 children.

Later, in 1900, another building was bought to create a charity house in Stará Turá. The house was re-built for the children by adding a room and its floors were changed. This institution retained the name Útulňa (Shelter) and as early as June 1901, several families were housed.

The mention of Kristína Roy in the November issue of the magazine Svetlo (1931) points to the spiritual dimension of this institution: *"On 10 August, former Juraj Mačica's house was consecrated for the abode Útulňa (the Shelter). It brought joy to our little family"*.

The children and their tutors, the deaconesses, lived as a big family. Children were regularly fed freshly cooked meals. On Sundays, children's church services were held and followed by the Sunday school with a focus on pastorally creating prerequisites in thinking for children's future life attitudes (trust in God, love of neighbour, abstinence, etc.).

In the afternoon, they took short walks in the Dubník grove where children would sing and play. To sum it up, their upbringing focused on all the physical, psychological, social, and spiritual needs of these children. From the beginning, Útulňa accommodated true orphans as well.

Door-to-door sellers' children started receiving proper care and their position improved significantly. Previously, many of them were left with large families, starving and malnourished. Given the family background of many children, e.g., alcoholic relatives, the Roy sisters focused not only on their nourishment, but also tried to protect them from the negative environmental influence. Their altruism drew from their familiarity with poverty and the families' difficult situations. They loved these children and did everything

to protect them from the social ills resulting from lack of proper upbringing in their family environment, and to reduce and prevent increased crime. Publicly, they battled alcoholism and tried to save children from it. These children were brought up under the healthy roof of "social pedagogy".

However, Útulňa was not dedicated only to door-to-door sellers' children—it was a kind of Christian social institute also attended by the townspeople who faced major social issues or lost their homes.

The positive experience with this diaconal work of the Roy sisters (and their collaborators) became a motivation even beyond the borders. Kristína Roy visited Kovačica (today in Serbia) in 1904, to inform about their work with orphans in Stará Turá. She inspired the Kovačica revivalists to follow her. The idea was so powerful that locals founded a small orphanage, and in 1939, they even built a grand two-storey building for social and charity work with orphans.

### 3.3.2. The Hospital and the First Slovak Evangelical Diaconia

According to the data, Stará Turá had about 6000 inhabitants at the time, many of them at solitary hillside homes. The physician worked in a single rented room without a waiting room down the street from Útulňa. Severely ill patients would be loaded onto hayracks with blankets, carried downtown to the doctor, and waited outside regardless of the season, then would move in front of the pharmacy to wait some more (at the time, the drugs were prepared ad hoc from powders). The nearest hospital was 50 km away.

The sight of people waiting for the physician was very common for Kristína Roy, because her window in Útulňa faced the street. These poor, ill people troubled her in her mercy. Although many tried to talk her out of it, in 1911 she decided a hospital must be built.

The locals mocked the sisters, calling them "sectarian women", but they overcame all obstacles and on 12 June 1911, building of the hospital started. The town provided 20,000 bricks. According to M. H. Moštenanová (1921, p. 23), many citizens, but mainly, those facing "Slovak poverty" worked to complete the massive project. It was a miracle given the era. Two physically weak women inspired an incredible effort to complete the building and succeeded. On 26 November 1911, a celebration was held—the hospital was consecrated and numerous guests from Tisovec, Nyiregyháza, Pressburg, Senica, Brezová, but also Yugoslavia (Kovačica, Kysáč) arrived.

On the ground floor of this nice two-floor building, the ambulance and staff rooms were located. On the upper floor, there were six smaller rooms with beds for patients, "the ward". However, the hospital did not have a doctor and the physician cooperating with the diaconia would come as needed, but at least once a week and gave the sisters advice on specific cases.

Therefore, Kristína Roy contacted Eva von Tiele Winckler, the founder of a large diaconical institute in Friedenshort, Silesia. For the first time, she visited Stará Turá in 1901 when Útulňa started working. For the second time, she returned when the diaconia was founded into which she put significant effort. According to Slezáčková (1991, p. 25), she invited two sisters from Stará Turá, Júlia Stanková and Anna Švejdová, to visit her diaconical institute in Friedenshort to study and become deaconesses.

On 12 May 1912, Eva von Tiele Winckler consecrated the deaconess' room in the hospital. Six deaconesses were ordained (including Kristína Roy who became the mother of the Stará Turá diaconia). She subsequently supported the diaconia by sending one of her best deaconesses, Alvina Hesse. She was German, but she soon learned Slovak and liked the local people very much. She became the head sister of the Slovak diaconia. The families in Stará Turá respected her and appreciated her assistance with birth and illnesses. Her selfless actions and behaviour invited respect and she was well liked. Her life can be characterised by selfless work in health care, which left her little time for herself. She served as a deaconess and nurse in the hospital for forty years. She was well trained in the field and doctors trusted her.

Deaconesses worked for free. If they received small gifts, they deposited them in the shared treasury. They lived exclusively to serve others and their service became a part of the town's life and improved it. However, this service was often very demanding.

In 1926, 30-year-old poet and translator Mária Rafajová from Brno travelled to Stará Turá and dedicated herself to social and charity work. She graduated from the Social College in Prague and became a Salvation Army officer. This Christian movement has improved the social situation in USA and Europe, including the territory of today's Czech and Slovak Republics. Rafajová was convinced to dedicate herself to this service when she saw that Kristína Roy not only wrote about love, but also really lived it. Thus, she decided to join this effort and permanently moved to Stará Turá. Kristína Roy found a successor in her and trained her to take over the diaconia. In 1930, Mária Rafajová initiated the separation of the diaconia from the Blue Cross; it was turned into an independent women's diaconical association named VIEROSLAVA. In Slovak, its etymological root is viera (faith) and sláva (glory), and it was inspired by Jesus' words addressed to Martha at Lazarus' grave: "Did I not tell you that if you *believe*, you will see the *glory* of God?" (John 11:40). However, the diaconia retained its original form from 1912.

In 1912, the diaconical institute in Stará Turá became the diaconical training centre in Slovakia. There were about 14–15 students at a time. After completing their education, the deaconesses travelled to Moravia, Czechia, and Silesia, or remained working in the Stará Turá institute.

According to sister Barbora Plesková, the two-year training comprised preparation in diaconical work basics. After ceremonially taking on the diaconical service, she started working in the hospital and in Útulňa like her fellow deaconesses. In Útulňa, there were only two rooms, but people were happy there and cherished their memories. It helped numerous people to recover. The elder Stará Turá inhabitants still remember cases when a patient was sent home to die from the Trenčín hospital, but recovered in Útulňa. The hospital was the first facility of this kind in the whole republic.

To support the Slovak diaconia, the Women's Association was founded. It attracted keen supporters, students, and helpers who left their previous professions and dedicated their lives to God's service and cared for the ill and abandoned. The deaconesses even wore uniforms. The auxiliary sisters or the so-called girl-friends also participated; they retained their jobs, but supported the deaconesses materially and with prayers and advice. These deaconesses ran the first Slovak diaconia in Stará Turá.

The records show that the deaconesses cared for the ill not only in the hospital, but also in local families. After completing their training, deaconesses went to work in other hospitals as well. They cared mainly for special patients who needed not only a qualified nurse, but also spiritual help in their old age and loneliness. Writer Terézia Vansová and even President T. G. Masaryk specifically requested the deaconesses to care for them.

### 3.3.3. Chalúpka (The Cottage)—The Children's Home

The children's home was Kristína Royová's dream. In 1925, the Royová sisters bought a house in Párovice and renovated the building of this unique social project in a year thanks to charitable donors. This second social institute, located in a cosy single-floor house with several rooms and a garden behind and not far from Útulňa (Shelter), was opened in 1926 (P. Farkaš 1950).

There were several rooms and another was added as a bedroom for the older children. Chalúpka accommodated 20–40 orphans who received parental love and care from the deaconesses. As far as is known, about 250 children were nurtured during its existence. These children, mainly orphans, grew up to be good and honourable people. It was even used by richer families who sent their problematic children for the summer to learn discipline. During WW2 after the Royová sisters both died, 10 Jewish children were sheltered here and saved from certain death.

There was an inscription above the door: "Behold, the God's house with people, and He will live with them!" Kristína Royová reminded the children: *"This is not our cottage, dear children! This is the house of Lord Jesus Christ and you are allowed to live here with Him"*.

Since the children's home had to be rebuilt later, almost all of Slovakia, Bohemia, and Moravia contributed financially to the construction of three rooms, the construction of a floor, the installation of a stove and electric lighting. People from places such as Romania, Hungary, Poland, etc. also donated.

The house and its renovation cost CZK 110,000. Only a CZK 27,000 debt was left at the time of handover. Immediately, the house began filling with children and supplies: smoked meat, eggs, butter, fresh and dried bread; the Royová sisters wove 20 m of canvas alone. These goods were divided between the two houses (Shelter and Cottage).

After the outbreak of WW2, the life in Stará Turá became harder, mainly for the Jews. Jewish parents would bring their children to Chalúpka for protection, but it was very dangerous for everyone, because the soldiers would often storm the house to search for hidden partisans.

### 3.4. The Vieroslava Association

This association housed deaconesses cooperating with Kristína Royová since 1913. In the *Večernica* magazine, she wrote the following on their goals: "*The goal of the Vieroslava association is to reduce people's suffering in Czechoslovakia, firstly, by caring for the very poor ill, bringing up orphans and neglected children, publishing Christian and temperance books and magazines, and training new sisters as deaconesses for this job*". Firstly, the association sent in their statute for approval to have the hospital and children's home included among social institutes in the Republic (VEČERNICA Magazine, 1931).

On 2 January 1930, a foundation meeting was held to approve the statute and name.

According to Slezáčková (1991, p. 45), the association respected general Christian and humanity principles: disseminate love and goodness, treat illnesses, and love children. The creed was Jesus' commandment: "Thou shalt love thy neighbour as thyself" and the activities were joined by both women and men. The name list of the contributors was published in the *Večernica* magazine. In the issue following the announcement, there were more than 200 contributors, and 7 years later, there were 506 contributors (including several men).

The Slovak association grew and members from Moravia and Czechia joined as well. In 1932, the Royová sisters were even sent on a mission to found a diaconal work in Yugoslavia.

### 3.5. Domov Bielych Hláv (The White Heads' Home)

The last home founded as part of the Royová sisters' social and charity work was the Domov Bielych Hláv. It too was built very quickly. In the *Večernica* magazine (published monthly by the Vieroslava women's association), Kristína Royová writes about the start of its construction. *"I have had two ideals in my life. To do good for the abandoned children and to sweeten the last days for the abandoned old people. The financial crisis has come. Reason warns me not to start now, but my withering strength calls: now or never"* (VEČERNICA Magazine, 1931).

Thus, the decision to start the construction was made and preparations ensued. Although the economic crisis arose at the time, the project entitled Zámoček (Little Castle) continued. The lot was located nearby Útulňa and all three buildings were close to one another in the upper part of the town. Thanks to *Večernica*, the believers, locals, and supporters abroad were asked to donate. Those who could not provide money were asked to pray for the project. Money and goods came in from the whole country. Mainly older people contributed, but the middle generation also helped. Two months after the call for donations was published in *Večernica*, Domov Bielych Hláv received large sums. Kristína Royová was an optimist, but difficulties with purchasing the land arose. They seemed impossible to overcome. However, Kristína Royová (71 at the time) endured; she would not stop. She prayed to God, fought, and won. The impossible became reality. In spring

1932, the construction started. After the rough construction had been completed, Kristína Royová thanked all the generous donors in the November issue of *Večernica* (1932, p. 175), and added a few thoughts about respect for old age: *"Our Domov Bielych Hláv already has its name on the facade. Many wonder about it, but we like the name. They say it should be simply called the old people's home. But it sounds sad. I disagree that people will meet in that house to die. There will be light before the evening comes. The sunset is as beautiful as the sunrise. The morning is nice, but the evening is cosy. If our house was to become a place of death and suffering, I would not even want to build it in the first place"*.

In 1932, the external facade was completed. Interior works started. On 1 May 1933, a ceremonial opening was held—an extraordinary event in Stará Turá. More than two thousand Slovaks, Czechs, Moravians, and Silesians attended. On 2 May 1933, Kristína Royová moved into the house as well and lived on the first floor for the final 3 years of her life. After Domov Bielych Hláv was opened, 25–30 new inhabitants moved in immediately.

Both sisters believed that faith in God comprised more than religiousness—it needed to be acted on. Kristína has summarised the legacy of her diaconate as follows: *"Your religion must be like the sun that gives off warmth. Thousands of people live around you and all of them need your love and help. God gives an opportunity to those who want to do good. Do not ask why your neighbour had bad luck! No! Ask how you can help them. If you have food, share it with the hungry. If you are strong, do a job for the weak and ill. If you have no children, care for the orphans. If you know the law, defend the widows. If you have a home, take in the homeless"* (*Svetlo*, June 1903:48).

## 4. Royová's Sisters as a Part of Women's, Volunteer, and International Roots of Slovak Social Work

Social work and the movement for its professionalisation had a tumultuous history. Recognition of professional social work faced numerous obstacles. The fact that social care providers were mostly women was important. Their unequal social position in the 19th century affected the perception of social work as a profession. Due to the inequality between women and men, the value, quality, and professionalism of social work performed by women was questioned. Social work is still perceived as a "women's job". However, understanding of the circumstances in which social work has been established may enhance our understanding of contemporary social work (Bosá 2013, p. 54).

Today's social work develops on the basis of three sources : the Christian tradition of love for one's neighbour and care, humanist philanthropy, and the women's movement—the subject of this study deals with all of them. Although the first source, Christian love and care, dominated in this case, it cannot be overlooked that in Stará Turá, at the time when women had few rights and kept silent in the public, two helpless, simple women with little education publicly called for a change. Although people disparaged them and called them heretics, the battalion of women speaks for itself and disseminates its legacy from under the Javorina Mountains around the many countries of the world. They proved that in the three pillars of social work, women played the key role. Women's movements such as the Vieroslava association (and Živena before 1869) "had enemies as well as keen supporters" (Votrubová 1931). In fact, it was women who volunteered to perform social work throughout the history of social work, as charity workers and deaconesses of the reformed churches.

Despite all misunderstandings, lack of money, and other obstacles, the Royová sisters succeeded. Their work represents one of the main motivations for social work: to invest in tomorrow. They invested in the lives of children whose parents had no time for them and could not see them for the whole year. They invested in children and youth to save them from abandonment and illiteracy and ensure they would not face difficulties in employment and society in general. They thought about the future. This kind of vision or motivation is absent in many social workers today who perceive social work as a mere job with fixed working time (M. H. Moštenanová 1921. Development and growth of Christian and social

work of Kristína Royová in Stará Turá. Ružomberok: Ján Párička Book Printing House in Ružomberok, 1921).

However, the social and charity work of the Royová sisters had a European, even global dimension; they travelled abroad to bring ideas back home and did not fear risk. They passionately tapped into the network of social and charity workers abroad. Slovakia became known around the world, even in some places in China. They helped orphans, the abandoned, and poor at home and abroad. They knew that a person is here for other people. Even people across the sea are like us and have the same needs, maybe even more urgent than in Slovakia. And the Royová sisters were conscious ambassadors of this idea, which is why they had so many followers. Instead of doing it for their own career, they did it for the well-being of the whole nation. In Kristína Royová's obituary, literary critic Ján Marták wrote, "*Kristína Royová has died. And her life was a rare one. Roses bloomed in her footsteps. Where the Lord sent her to serve, broken, withered and trampled flowers grew again. Her life was full of faith and love. Only a true, deep, and bottomless faith—Christian love that can carry mountains can create such a special person, able to bring such blessed fruit. No good work in the world disappears without fruit*". (NÁRODNÉ NOVINY, 1937, Obituary—critic Ján Marták.).

When looking for appropriate approaches to solving the pastoral, diaconical, and social problems of today, it is appropriate that we pay attention to our ancestors' legacies.

*Politically Forced Termination of Activity*

The onset of communist totality in Czechoslovakia in the years after WWII (1948) significantly limited the work founded by the Royová sisters. The Blue Cross was cancelled and the revival movement dissolved into a number of Protestant denominations (i.e., remained in the Evangelical church). The newly created atheistic communist state gradually cancelled the individual diaconical areas, because the Christian basis of the diaconal work was considered as an enemy of the official line of the state. It also concerned Chalúpka—a children's home. The last Jewish children were there until 1948, when their parents or other relatives came to take them into the care of their families. Only the Evangelical orphans from the region remained and later they were assigned to different state children's homes across Slovakia, because the state aimed to weaken their previous religious education.

## 5. Conclusions

We observed the following in the examined material:

The Royová sisters had the ability to recognise problems. Instead of seeing faith as an empty ritual or cold dogma, they lived it. They acted on it. They shared what they saw in their books and songs, and they reflected on it in their social service provided to others. They could not pretend to be blind, deaf, or indifferent. It is valuable that, due to their love of God and people, their spirituality did not deform into a negativist attitude towards the environment. On the contrary, it motivated them to act in favor of people who, due to insufficient upbringing and education, poverty, and alcoholism, needed help. In order to determine what is normal and what requires change, the existence of corresponding concepts is necessary and not just an assessment of what is common at a given time. Neglected children, poverty, and adult alcoholism were common in their time. Although this represented a "statistical normal", the Royová sisters did not see it as a "functional normal", that is, a state of how one could live under the existing conditions.

Searching and finding effective ways to manage change gradually resurfaced. They did not stop at the theoretical conclusion that the problems could be changed, but the Royová sisters put their concepts into practice and thus positively influenced the lives of many people.

The complexity of the approach in their work is valuable. They did not focus selectively on only a selected part—e.g., the spiritual level (which is represented by their religious activity and literary and song creation) or for diaconal (social) work only. The Royová sisters were looking for options for comprehensive support and care. They worked on the development of prerequisites for a change in thinking and a deepening of spiritual

life (Christian literature), and cooperated with temperance associations (Blue Cross). By doing so, they realised the prevention of the negative effects of alcoholism and created prerequisites for improving the social conditions of the participating families. Finally, it is also valuable that they provided diaconal assistance for orphans, neglected children, and the elderly, dependent on help, in the social facilities they established for this purpose.

The beginnings of the institutionalisation of social and charitable work in Slovakia and Serbia occurred through the organisations founded by the Royová sisters. These institutions and associations included, in particular, the Sunday school, the Blue Cross, the Shelter, the first Slovak Evangelical Diaconate, the children's orphanage, the Vieroslava Association, the Home of White Heads, all of them in Stará Turá, Slovakia, and the orphanage in Kovačica, Serbia.

Christian-social interpersonal contribution of the Royová sisters to the development of Slovak and European social work was personified by their cooperation with several personalities of social and charitable work at the international, national, and local levels. These personalities included, for example, Alica Masaryková, Eva von Tile Winckler, Alvina Hesse, Borka Plesková, Mária Rafajová, Ján Chorvát, and many others.

The contribution of the Royová sisters was significant in creating women volunteer and international roots of social and charitable work in Slovakia and in Europe. Primarily, their contribution to the women's social work movement, their involvement as volunteers, and the strong international dimension of the Royová Sisters' social and charitable activities consisted not only in activities throughout their lives but also in their ever-living legacy, not only in Europe but also on other continents.

We see one of the sources of this rich set of activities in the fact that Kristína Royová was strongly rooted in a living Christian faith; she had a clearly defined religious and national identity as well as ordered values of her being. Her work shows that she felt "loved by God". Therefore, she had her self-understanding and self-worth built from within, and in this sense, she was not dependent on the reaction of the external environment.

Similarly, this can be observed with her sister Maria. Their self-esteem could not be damaged by the disparagement by some townspeople who considered them sectarians. Because they had a healthy self-image, they did not need to waste energy on unnecessary disputes. A rich inner life became an important prerequisite for the development of their ministry. This was the engine of their strong commitment and the source of their selfless approach. Their work was successful because they considered service to target groups (children, sick, seniors) and not their profit to be of high value.

This poses a critical question of motivation for today's workers in professional positions in the field of social work and deaconry, but also pastoral care, whether these workers enter these services for the purpose of solving their own problems or work there for the benefit of target groups. At the same time, it also poses a critical question to the founders of these positions, because the actors of these services also have their own lives and costs. Therefore, it is appropriate that not only the needs of clients but also those who help them cope with the difficulties of human existence are responsibly covered.

With a detailed look at the examined material, we can add other findings:

Reach. In their activity, we can also recognise a significant pastoral dimension, which is represented, for example, by the fact that Kristína Royová reminded disadvantaged children who were in care at Chalúpka, "This is not our cottage, dear children! This is the cottage of the Lord Jesus, and you may live in it with Him!" In this way, she was passing on to them her life attitude of faith, the awareness of privilege, which became the basis of their positive self-acceptance and self-image, and the basis for trust in God in the future as protection against short-cut reactions in times of crisis situations. This was done in the context of daily loving material care.

Resources. Naturally, the stated range of activities was demanding on material and personnel resources. Credible and meaningful activity appealed to people who were willing to support it financially; thus, for many, it became an opportunity for their involvement in doing good. Personnel resources required the transfer of spiritual motivation for this work,

but also qualification preparation. That is why it is remarkable that the activity, which was first dependent on foreign qualification support, later became a source of professional training not only for securing one's own activity, but also for the development of services in other locations.

The authors hope that this modest historical research will slightly raise awareness of these valuable people and their work. There are many social projects of high quality these days, but in the context of the end of the 19th century and beginning of the 20th century, the described activities represented impressive steps. Their quality is also referred to by their contemporary doctor Ladislav Minárik, who remembers them as follows: "I personally met the loving and selfless deaconesses in Stará Turá when I worked as a medic in Uhor's partisan group during the Slovak National Uprising (WWII). After returning from hiding in Vetešovský jarok, we found a shelter in Stará Turá, in the deaconesses' hospital; for more than a week they would take care of us, fed us, bathed us, removed lice from us, gave us clean clothes, and never asked about our religion. Their humane actions deserve to be pointed out and never to be forgotten".

Christian diaconia, through love for God, opened their eyes to be sensitive to the needs of others and supported their faith that it is possible to find appropriate and effective solutions that they then implemented into life. This can support our motivation for the creative search for effective forms of helping dependent people even in today's multi-crisis environment.

**Author Contributions:** "Historical resources; formal analysis"—J.N.; "Methodology, Aspects of social work"—P.J.; "Theological anthropology and spiritual aspects; writing—review and editing,"—A.M. All authors have read and agreed to the published version of the manuscript.

**Funding:** This research received no external funding.

**Institutional Review Board Statement:** Not applicable.

**Informed Consent Statement:** Not applicable.

**Data Availability Statement:** Research data are available in archive resources by Anna Chrťanová, Kysáč, Serbia and in The Roy Sisters Museum, Stará Turá, Slovakia.

**Acknowledgments:** The authors thank Anna Chrťanová from Kysáč, Serbia, for providing letters, photographs, records, and manuscripts from her personal archive.

**Conflicts of Interest:** The authors declare no conflict of interest.

## Note

[1] Ján Chorvát initiated the founding of Blue Cross branches across the Kingdom of Hungary (Tisovec, Novohrad, Šariš, Gemer, Mengusovce, Ozdín), and other branches were founded in Yugoslavia (Kysáč, Kovačica, Petrovec) as well.

## References

### Archive Resources

Časopis SVETLO (1903; 1931). [SVETLO Magazine (1903; 1931)].
Časopis VEČERNICA (1925–1933) [VEČERNICA Magazine (1925–1933)].
NÁRODNÉ NOVINY, 1937, Nekrológ—Kritik Ján Marták (1937). [NÁRODNÉ NOVINY, 1937, Obituary-critic Ján Marták].
Prvé vydanie novely Kristíny Royovej Bez Boha na svete u Jána Bežu v Senici (1893). [The first edition of Kristína Roy's short story Without God in the World by Ján Beža in Senica (1893)].

### Secondary Resources

Bosá, Monika. 2013. *Feministické Korene Sociálnej Práce [Feminist Roots of Social Work]*. Prešov: Filozofická fakulta Prešovskej univerzity.
Brnula, Peter. 2013. *Sociálna Práca—Dejiny, Teórie a Metódy [Social Work—History, Theories and Methods]*. Bratislava: IRIS.
Farkaš, Pavel. 1950. *Úryvky z histórie nášho sirotinca [Excerpts from the History of Our Orphanage]*. Kovačica: Ev. a.v. cirkevný zbor.
Fischer, Bohumil. 1901. *Modrý kríž*. Viedeň: Knihovňa Modrého kríža.
Global Definition of Social Work. 2014. Available online: https://www.ifsw.org/what-is-social-work/global-definition-of-social-work/ (accessed on 7 July 2021).
Hanes, Pavel. 2022. Theological Axiology of Reality. *The Person and the Challenges* 12: 19–36. [CrossRef]

Hanes, Pavel, and Dana Hanesová. 1999. Matej Bel—pedagóg. [Matej Bel—An Educator]. In *Odkaz Maeja Bela generáciám XXI. storočia. [The Legacy of Matej Bel to Generations XXI. Century]*. Edited by Ondrej Lupták. Banská Bystrica: Univerzita Mateja Bela, Pedagogická fakulta, pp. 13–31.

Jakš, Milan, and Samuel Spišiak. 2006. *Život a Dielo Sestier Royových [The Life and Work of the Roy Sisters]*. Bratislava: MSEJK.

Kováčiková, Dagmar. 2000. *Základné Otázky Dejín Sociálnej Práce [Fundamental Questions of the History of Social Work]*. Žilina: Žilinská univerzita—EDIS.

Lupták, Ondrej. 1999. Úvod. [Introduction]. In *Odkaz Maeja Bela Generáciám XXI. storočia. [The Legacy of Matej Bel to Generations XXI. Century]*. Banská Bystrica: Univerzita Mateja Bela, Pedagogická fakulta, pp. 5–12.

Moštenanová, Miloslava Hermína. 1921. *Vývin a Vzrast Kresťanskej a Sociálnej Práce Kristíny Royovej na Starej Turej [The Development and Growth of Kristína Royova's Christian and Social Work in Stará Turá]*. Ružomberok: Kníhtlačiareň Jána Páričku v Ružomberku.

Podolan, Peter, and Miriam Viršanská. 2014. *Slovenské dejiny III. 1780–1914*. Bratislava: Literárne Informačné Centrum.

Slavka, M. 2007. *Staroturanská diakonia. [Diaconia of Stara Tura]*. Stará Turá: Evanjelický a.v. cirkevný zbor na Starej Turej.

Slezáčková, J. 1991. *Život a dielo sestier Márie a Kristíny Royových. [The life and work of sisters Maria and Kristina Royova]*. Stará Turá: Evanjelický a.v. cirkevný zbor na Starej Turej.

*Spevník—400 kresťanských piesní.* 2014. Bratislava: Bratská Jednota Baptistov v SR.

Votrubová, Š. 1931. *ŽIVENA—Jej Osudy a Práca. [ŽIVENA—Its Destinies and Work]*. Martin: ŽIVENA, Spolok slovenských žien.

