# Peer review of "The Diaconal Work of Sisters Kristína and Mária Royová—An Example of the Link between Christian Anthropology and Social Work"

_religions, doi:10.3390/rel15010009_

Round 1
Reviewer 1 Report
Comments and Suggestions for Authors
It is fascinating to learn about the history of the Evangelical Awakening and Revival in Slovakia by way of the work of the Royová sisters. The article here represents an important document for Slovakian Church history as much as a thoughtful reflection for consideration of dialogue for the Catholic Church to think about new ministries as a form of synodality. This indeed could be quite challenging especially at looking at the notion of truth in love (Eph 4:15), that is to say, discovering ways of love through good ministry.
Interestingly as well, the article has value in terms of giving time and space to reflect on the importance and history of women's ministry. Looking at types of charity like "disseminate love and goodness, treat illnesses, and love children" speaks of maternity which is only a step away from compassion and mercy - a whole theology of "investing in tomorrow".
In terms of improving the article, I suggest more thorough theological engagement with the themes of Christian anthropology and pastoral care. I suggest for example highlighting and developing with research (not just personal reflection) some important themes in relation to Christian anthropology like covenant (relationship with God), being ethically oriented and transcendence (openness to God's will and spirit to be a person-in-Christ). Also, try to think further how pastoral care differs from social work. This will give more force to the careful research done.
Author Response
Thank you for your review report, please see the attachment.

Reviewer 2 Report
Comments and Suggestions for Authors
I do not see any problems.
Author Response

(The authors gave the same response as above.)

Reviewer 3 Report
Comments and Suggestions for Authors
The article documents an interesting fragment of the Church'as activities (church people). Saved from oblivion. The article proves good scientific competences.
Author Response

(The authors gave the same response as above.)
